# Neural Architecture Generator Optimization

**Binxin Ru**
Machine Learning Research Group
University of Oxford, UK
robin@robots.ox.ac.uk

**Pedro M. Esperança**
Huawei Noah's Ark Lab
London, UK
pedro.esperanca@huawei.com

**Fabio M. Carlucci**
Huawei Noah's Ark Lab
London, UK
fabio.maria.carlucci@huawei.com

## Abstract

Neural Architecture Search (NAS) was first proposed to achieve state-of-the-art performance through the discovery of new architecture patterns, without human intervention. An over-reliance on expert knowledge in the search space design has however led to increased performance (local optima) without significant architectural breakthroughs, thus preventing truly novel solutions from being reached. In this work we 1) are the first to investigate casting NAS as a problem of finding the optimal network generator and 2) we propose a new, hierarchical and graph-based search space capable of representing an extremely large variety of network types, yet only requiring few continuous hyper-parameters. This greatly reduces the dimensionality of the problem, enabling the effective use of Bayesian Optimisation as a search strategy. At the same time, we expand the range of valid architectures, motivating a multi-objective learning approach. We demonstrate the effectiveness of this strategy on six benchmark datasets and show that our search space generates extremely lightweight yet highly competitive models. The code is available at https://github.com/rubinxin/vega_NAGO.

## 1 Introduction

Neural Architecture Search (NAS) has the potential to discover paradigm-changing architectures with state-of-the-art performance, and at the same time removes the need for a human expert in the network design process. While significant improvements have been recently achieved [1, 2, 3, 4, 5, 6], this has taught us little about *why* a specific architecture is more suited for a given dataset. Similarly, no conceptually new architecture structure has emerged from NAS works. We attribute this to two main issues: (i) reliance on over-engineered search spaces and (ii) the inherent difficulty in analyzing complex architectures.

The first point is investigated in [7]. In order to reduce search time, current NAS methods often restrict the macro-structure and search only the micro-structure at the cell level, focusing on which operations to choose but fixing the global wiring pattern [1, 8, 9, 10]. This leads to high accuracy but restricts the search to local minima: indeed deep learning success stories, such as ResNet [11], DenseNet [12] and Inception [13] all rely on specific global wiring rather than specific operations.

The second issue appears hard to solve, as analyzing the structure of complex networks is itself a demanding task for which few tools are available. We suggest that by moving the focus towards *network generators* we can obtain a much more informative solution, as the whole network can then be represented by a small set of parameters. This idea, first introduced by [14], offers many

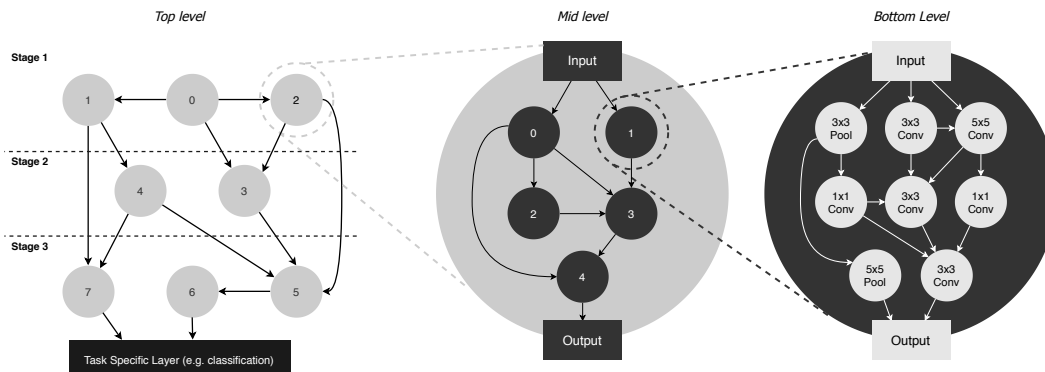

Figure 1: Architecture sampled from HNAG, given hyperparameters $\Theta$. Each node, both in the top-level and mid-level graphs, is an independently sampled graph. Finally, at the bottom level each node corresponds to an independently sampled atomic operation. Note how features at the top level can flow between different stages (e.g. from node 1 and 4 to 7), which is beneficial for certain tasks.

advantages for NAS: the smaller number of parameters is easier to optimize and easier to interpret when compared to the popular categorical, high-dimensional search spaces. Furthermore it allows the algorithm to focus on macro differences (e.g. global connectivity) rather than the micro differences arising from minor variations with little impact on the final accuracy.

To summarize, our main contributions are as follows.

1) A **Network Architecture Generator Optimization framework** (NAGO), which redirects the focus of NAS from optimizing a single architecture to optimizing an architecture generator. To the best of our knowledge, we are the first to investigate this direction and we demonstrate the usefulness of this by using Bayesian Optimization (BO) in both multi-fidelity and multi-objective settings.

2) A new **hierarchical, graph-based search space**, together with a stochastic network generator which can output an extremely wide range of previously unseen networks in terms of wiring complexity, memory usage and training time.

3) **Extensive empirical evaluation** showing that NAGO achieves state-of-the-art NAS results on a variety of vision tasks, and finds lightweight yet competitive architectures.

## 2   Neural Architecture Generator

Previous research has shown that small perturbations in the network's structure do not significantly change its performance, i.e. the specific connection between any single pair of nodes is less important than the overall connectivity [14, 7]. As such, we hypothesize, and experimentally confirm in Section 4.1, that architectures sampled from the same generative distribution perform similarly. This assumption allows us to greatly simplify the search and explore more configurations in the search space by only evaluating those sampled from different generator hyperparameters. Therefore, instead of optimizing a specific architecture, we focus on finding the optimal hyperparameters for a stochastic network generator [14].

### 2.1   Hierarchical Graph-based Search Space (HNAG)

Our network search space is modelled as a hierarchical graph with three levels (Figure 1). At the top-level, we have a graph of cells. Each cell is itself represented by a mid-level graph. Similarly, each node in a cell is a graph of basic operations (conv3×3, conv5×5, etc.). This results in 3 sets of graph hyperparameters: $\boldsymbol{\theta}_{top}, \boldsymbol{\theta}_{mid}, \boldsymbol{\theta}_{bottom}$, each of which independently defines the graph generation model in each level. Following [14] we use the Watts-Strogatz (WS) model as the random graph generator for the top and bottom levels, with hyperparameters $\boldsymbol{\theta}_{top} = [N_t, K_t, P_t]$ and $\boldsymbol{\theta}_{bottom} = [N_b, K_b, P_b]$; and use the Erdős–Rényi (ER) graph generator for the middle level, with hyperparameters $\boldsymbol{\theta}_{mid} = [N_m, P_m]$ to allow for the single-node case [1]. This gives us the flexibility to reduce our search space to two levels (when the mid-layer becomes single node) and represent a

DARTS-like architecture. Indeed HNAG is designed to be able to emulate existing search spaces while also exploring potentially better/broader ones.

By varying the graph generator hyperparameters and thus the connectivity properties at each level, we can produce a extremely diverse range of architectures (see end of this section). For instance, if the top-level graph has 20 nodes arranged in a feed-forward configuration and the mid-level graph has a single node, then we obtain networks similar to those sampled from the DARTS search space [1]. While if we fix the top-level graph to 3 nodes, the middle level to 1 and the bottom-level graph to 32, we can reproduce the search space from [14].

**Stages.** CNNs are traditionally divided into stages, each having a different image resolution and number of channels [15, 1, 14]. In previous works, both the length of each stage and the number of channels were fixed. Our search space is the first that permits the learning of the optimal channel ratio as well as the channel multiplier for each stage. To do so, we define two hyperparameter vectors: stage ratio $\boldsymbol{\theta}_S$ and channel ratio $\boldsymbol{\theta}_C$. $\boldsymbol{\theta}_S$ is normalized and represents the relative length of each stage. For example, if there are 20 nodes at the top level and $\boldsymbol{\theta}_S = [0.2, 0.2, 0.6]$ then the three stages will have 4, 4 and 12 nodes, respectively. $\boldsymbol{\theta}_C$ controls the number of channels in each stage; e.g. if it is set to $[4, 1, 4]$ than stages 1 and 3 hold the same number of channels while stage 2 only holds one fourth of that. The absolute number of channels depends on the overall desired number of parameters while $\boldsymbol{\theta}_C$ only controls the relative ratio.

**Merging options and Operations.** When multiple edges enter the same node, they are merged. Firstly, all activations are downscaled via pooling to match the resolution of the smallest tensor. Note we only tried pooling for our work but strided convolution is an alternative option to achieve the same effect. Likewise, we use $1 \times 1$ convolutions to ensure that all inputs share the same number of channels. Then, independently for each node, we sample, according to the probability weights $\boldsymbol{\theta}_M$, one merging strategy from: weighted sum, attention weighted sum, concatenation. Each atomic operation is sampled from a categorical distribution parameterized with $\boldsymbol{\theta}_{op}$, which can be task specific.

Therefore, our search space is fully specified by the hyperparameters $\boldsymbol{\Theta} = [\boldsymbol{\theta}_{top}, \boldsymbol{\theta}_{mid}, \boldsymbol{\theta}_{bottom}, \boldsymbol{\theta}_S, \boldsymbol{\theta}_C, \boldsymbol{\theta}_M, \boldsymbol{\theta}_{op}]$. The top-level enables a mixture of short- and long-range connections among features of different stages (resolutions/channels). The mid-level regulates the search complexity of the bottom-level graph by connecting features computed locally (within each mid-level node).[2] This serves a function similar to the use of cells in other NAS method but relaxes the requirement of equal cells. Our hierarchical search expresses a wide variety of networks (see Section 4.1). The total number of networks in our search space is larger than $4.58 \times 10^{56}$. For reference, in the DARTS search space that number is $8^{14} \approx 4.40 \times 10^{12}$ (details in Appendix A).

## 2.2 BO-based Search Strategy

Our proposed hierarchical graph-based search space allows us to represent a wide variety of neural architectures with a small number of continuous hyperparameters, making NAS amenable to a wide range of powerful BO methods such as multi-fidelity and multi-objective BO. The general algorithm for applying BO to our search space is presented in Appendix B.

**Multi-fidelity BO (BOHB).** We use the multi-fidelity BOHB approach [16], which uses partial evaluations with smaller-than-full budget in order to exclude bad configurations early in the search process, thus saving resources to evaluate more promising configurations and speeding up optimisation. Given the same time constraint, BOHB evaluates many more configurations than conventional BO which evaluates all configurations with full budget.

**Multi-objective BO (MOBO).** We use MOBO to optimize for multiple objectives which are conflicting in nature. For example, we may want to find architectures which give high accuracy but require low memory. Given the competing nature of the multiple objectives, we adapt a state-of-the-art MOBO method to learn the Pareto front [17] [3]. The method constructs multiple acquisition functions, one for each objective function, and then recommends the next query point by sampling the point with

the highest uncertainty on the Pareto front of all the acquisition functions. We modify the approach in the following two aspects for our application:

1) *Heteroscedastic surrogate model.* We use a stochastic gradient Hamiltonian Monte Carlo (SGHMC) BNN [18] as the surrogate, which does a more Bayesian treatment of the weights and thus gives better-calibrated uncertainty estimates than other alternatives in prior BO-NAS works [19, 20, 21]. SGHMC BNN in [18] assumes homoscedastic aleatoric noise with zero mean and constant variance $w_n^2$. By sampling the network weights $w_f$ and the noise parameter $w_n$ from their posterior $w^i \sim p(w|D)$ where $w = [w_f, w_n]$ and $D$ is the surrogate training data, the predictive posterior mean $\mu(f|\boldsymbol{\Theta}, D)$ and variance $\sigma^2(f|\boldsymbol{\Theta}, D)$ are approximated as:

$$\mu(f|\boldsymbol{\Theta}, D) = \frac{1}{N} \sum_{i=1}^{N} \hat{f}(\boldsymbol{\Theta}; w_f^i), \qquad \sigma^2(f|\boldsymbol{\Theta}, D) = \frac{1}{N} \sum_{i=1}^{N} \hat{f}(\boldsymbol{\Theta}; w_f^i)^2 - \mu(f|D)^2 + w_n^2 \quad (1)$$

However, our optimization problem has heteroscedastic aleatoric noise: the variance in the network performance, in terms of test accuracy or other objectives, changes with the generator hyperparameters (Figure 2). Therefore, we propose to append a second output to the surrogate network and model the noise variance as a deterministic function of the inputs, $w_n^2(\boldsymbol{\Theta})$. Our heteroscedastic BNN has the same predictive posterior mean as Equation (2.2) but a slightly different predictive posterior variance: $\sigma^2(f|\boldsymbol{\Theta}, D) = \frac{1}{N} \sum_{i=1}^{N} \left( \hat{f}(\boldsymbol{\Theta}; w_f^i)^2 + \left(w_n^i(\boldsymbol{\Theta})\right)^2 \right) - \mu(f|\boldsymbol{\Theta}, D)^2$. Our resultant surrogate network comprises 3 fully-connected layers, each with 10 neurons, and two outputs. The hyperparameter details for our BNN surrogate is described in Appendix C.

We verify the modelling performance of our heteroscedastic surrogate network by comparing it to its homoscedastic counterpart. We randomly sample 150 points from BOHB query data for each of the five image datasets and randomly split them into a train-test ratio of 1:2. The median results on negative log likelihood (NLL) and root mean square error (RMSE) over 10 random splits are shown in Table 1. The heteroscedastic model not only improves over the homoscedastic model on RMSE, which depends on the predictive posterior mean only, but more importantly, shows much lower NLL, which de-

Table 1: Regression performance of heteroscedastic (Het) and homoscedastic (Hom) BNN surrogates, trained on 50 generator samples and tested on 100 samples, in terms of negative log-likelood (NLL) and root mean square error (RMSE)

|  | NLL | | RMSE | |
|---|---|---|---|---|
|  | Hom | Het | Hom | Het |
| CIFAR10 | 5.92 | **3.43** | 0.02 | **0.01** |
| CIFAR100 | 7.15 | **0.89** | 0.02 | **0.02** |
| SPORT8 | 23.8 | **19.0** | 0.15 | **0.14** |
| MIT67 | 7.23 | **-0.92** | 0.12 | **0.11** |
| FLOWERS102 | 15.6 | **7.49** | 0.19 | **0.18** |

pends on both the predictive posterior mean and variance. This shows that the heteroscedastic surrogate can model the variance of the objective function better, which is important for the BO exploration.

2) *Parallel evaluations per BO iteration.* The original multi-objective BO algorithm is sequential (i.e. recommends one new configuration per iteration). We modify the method to a batch algorithm which recommends multiple new configurations per iteration and enjoys faster convergence in terms of BO iterations [22, 23]. This allows the use of parallel computation to evaluate batches of configurations simultaneously. We collect the batch by applying the local penalisation on the uncertainty metric of the original multi-objective BO [23]. See Appendix D for details.

# 3 Related Work

**Neural Architecture Search (NAS).** NAS aims to automate the design of deep neural networks and was initially formulated as a search problem over the possible operations (e.g. convolutions and pooling filters) in a graph. Several approaches have been proposed that outperform human-designed networks in vision tasks: reinforcement learning [24, 3], evolution [25], gradient descent [1, 8] and multi-agent learning [4]. To achieve a computationally feasible solution, these works rely on a manually-designed, cell-based search space where the macro-structure of the network (the global *wiring*) is fixed and only the micro-structure (the *operations*) is searched. Some recent works also start to look into the search space design [26, 27] but study very different perspectives from our work.

**Network Wiring.** Recent works have explored the importance of the wiring of a neural network. [28] use a gradient-based approach to search the network wiring via learnable weights, and [29] search for the wirings among channels instead of layers; both modify existing network architectures instead of discovering new ones. In contrast, [14] build networks from random graph models.

The concept of stochastic network generator was introduced by [14] who show that networks based on simple random graph models (RNAG) are competitive with human and algorithm designed architectures on the ImageNet benchmark. In contrast with our work, they do not offer a strategy to optimize their generators. The second main difference with their work lies in our search space (HNAG). While RNAG is *flat* with 3 sequentially connected graphs, HNAG is *hierarchical* with each node in the higher level corresponding to a graph in the level below, leading to a variable number of nested graphs. Our 3-level hierarchical structure is not only a generalization which enables the creation of more complex architectures, but it also allows the creation of local clusters of operation units, which result in more memory efficient models (Fig.3) or architectures with fewer nodes (Sec.4.3). Moreover, the HNAG top level provides diverse connections across different stages, leading to more flexible information flow than RNAG. Finally, nodes in RNAG only process features of fixed channel size and resolution within a stage while those in HNAG receive features with different channel sizes and resolution. To summarize, our work distinguishes itself by proposing a significantly different search space *and* a BO based framework to optimize it, empirically evaluating it on 6 datasets and further investigating the multi-objective setting [6, 30].

**BO for NAS.** BO has been widely used for optimizing the hyperparamters of machine learning algorithms [31, 32, 33, 34, 35, 16, 36] and more recently it has found applications in NAS [37, 38, 19, 20, 21]. However, current NAS search spaces—even cell-based ones—are noncontinuous and relatively high dimensional [39], thus unfavourable to conventional BO algorithms that focus on low-dimensional continuous problems [40, 41]. Our proposed HNAG which addresses the above issues can make NAS amenable to BO methods.

**Hierarchical Search Space.** [42] proposes to search for the outer network-level structure in addition to the commonly searched cell-level structure; [43] proposes a factorised hierarchical search space which permits different operations and connections in different cells; [44] introduce a hierarchical construction of networks with higher-level motifs being formed by using lower-level motifs. However, none of these prior works propose to optimise architecture generators instead of single architectures, which is the main contribution of our search space; Such generator-based search space formulation leads to advantages like high expressiveness, representational compactness and stochasticity.

# 4 Experiments

We experiment with the following two network generators.

1) Hierarchical Neural Architecture Generator (**HNAG**): our proposed 3-level, hierarchical generator. Due to resource constraint, we limit our search space to the 8 *random graph generator hyperparameters* $[\boldsymbol{\theta}_{top}, \boldsymbol{\theta}_{mid}, \boldsymbol{\theta}_{bottom}]^4$. The search ranges of these hyperparameters are in Appendix E. Following [14] we fix $\boldsymbol{\theta}_S = [0.33, 0.33, 0.33]$ and $\boldsymbol{\theta}_C = [1 : 2 : 4]$. Experiments on expanded search spaces are shown in Appendix H. The absolute number of channels for each stage is computed by multiplying the channel ratio with a constant which is calculated based on our parameter limit.

2) Randomly Wired Neural Architecture Generator (**RNAG**): the flat network generator in [14] which connects three WS graphs in sequence to form an architecture. We optimize the three WS hyperparameters $(N, K, P)$ for each stage, leading to 9 hyperparameters in total.

For both HNAG and RNAG, we apply multi-fidelity and multi-objective BO to optimize their hyperparameters, leading to 4 methods for comparison: HNAG-BOHB, HNAG-MOBO, RNAG-BOHB, RNAG-MOBO. To verify the effectiveness of our search strategy as well as expressiveness, we also evaluate the performance of random samples drawn from HNAG (HNAG-RS) and use it as another baseline. For all generator-based methods, we use summation to merge the inputs and only use 3×3 convolution as the operation choice, unless otherwise stated.

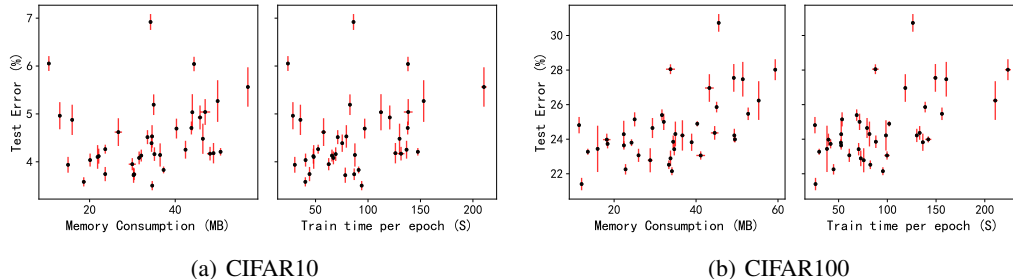

| (a) CIFAR10 | (b) CIFAR100 |

Figure 2: Expressiveness of our HNAG search space. The above plots shows the mean and standard deviation of test error vs. memory consumption and training time per epoch achieved by 40 random generator hyperparameters for CIFAR10 and CIFAR100. The mean and standard deviation of results over the 8 sampled architectures for each generator hyperparameter are presented.

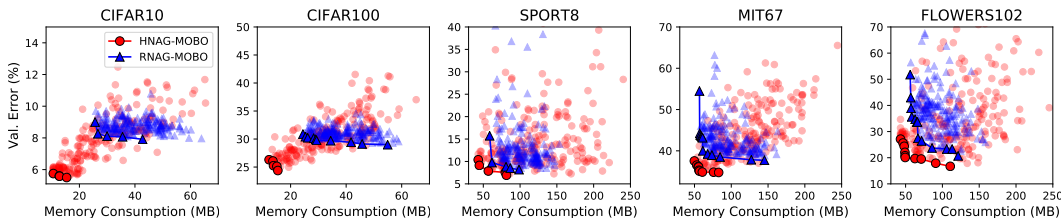

Figure 3: Query data by MOBO for HNAG (red) and RNAG (blue). The Pareto optimal configurations are highlighted in solid lines and filled markers. Each BO evaluation uses 200 training epochs for SPORT8 and 60 training epochs for the other datasets.

**Datasets.** We perform experiments on a variety of image datasets: CIFAR10, CIFAR100 [45], IMAGENET [46] for object recognition; SPORT8 for action recognition [47]; MIT67 for scene recognition [48]; FLOWERS102 for fine-grained object recognition [49]. We limit the number of network parameters to $4.0$M for small-image tasks and $6.0$M for large-image tasks.

**Complete training protocol.** For all datasets except IMAGENET, we evaluate the performance of the (Pareto) optimal generators recommended by BO by sampling 8 networks from the generator and training them to completion (600 epochs) with initial learning rate $0.025$ and batch size 96. For IMAGENET, we follow the complete training protocol of small model regime in [14], which trains the networks for 250 epochs with an initial learning rate of 0.1 and a batch size of 256. We use cutout with length 16 for small-image tasks and size 112 for large-image tasks. Note that we do not use DropPath or other advanced training augmentations. All experiments use NVIDIA Tesla V100 GPUs.

## 4.1 Expressiveness of the search space

We evaluate the performance of $40$ randomly sampled network generator hyperparameters for CIFAR10 and CIFAR100 (Figure 2). The performance of each hyperparameter set is evaluated by training 8 neural network architectures generated by following the complete training protocol (described above) and evaluating on the test datasets. Similar plots following the training protocol in the BO search phase are presented in Appendix G. There are three observations we highlight.

Firstly, the accuracy (test error) and efficiency (memory consumption[5] or training time) achieved by different generator hyperparameters are spread over a considerable range. This shows that our search space can express a variety of generators whose networks have different characteristics, thus optimization is meaningful. Secondly, the networks produced by good generator hyperparameters mostly have small variation in their accuracy and memory, which justifies our proposal to focus on the generator instead of the network. It also supports our approach to assess the performance of a specific generator configuration with only one architecture sample in our BO phase. Third, there exist Pareto optimal generator hyperparameters in our search space as some of them result in architectures which are both efficient and accurate. This justifies our motivation for performing MOBO.

Table 2: Validation accuracy (%) and search cost (GPU days) for BOHB results. The accuracy reported is obtained in the BOHB search setting which uses large batch sizes based on GPU machine memory and trains the network sample for 400 epochs for SPORT8 and 120 epochs for the other datasets.

|  | RNAG-BOHB | | HNAG-BOHB | |
|---|---|---|---|---|
|  | Accuracy | Cost | Accuracy | Cost |
| CIFAR10 | 93.5 | 14.0 | **95.6** | **12.8** |
| CIFAR100 | 72.2 | 11.8 | **77.2** | **10.4** |
| SPORT8 | 94.7 | 23.4 | **95.3** | **17.6** |
| MIT67 | 67.7 | 22.6 | **71.8** | **20.0** |
| FLOWERS102 | 91.4 | 11.2 | **93.3** | **10.6** |

Table 3: The optimal values found for the 8 generator hyperparameters for single-objective BOHB (top block) and MOBO (bottom block) experiments. "MPS" is Memory Per Sample.

|  | Top | | | Mid | | Bottom | | | MPS |
|---|---|---|---|---|---|---|---|---|---|
|  | N | K | P | N | P | N | K | P | (MB) |
| CIFAR10 | 8 | 5 | 0.6 | 1 | 0.7 | 5 | 4 | 0.2 | 17 |
| CIFAR100 | 8 | 5 | 0.4 | 1 | 0.7 | 4 | 2 | 0.4 | 14 |
| SPORT8 | 7 | 2 | 0.9 | 5 | 0.8 | 6 | 3 | 0.6 | 121 |
| MIT67 | 9 | 4 | 0.6 | 1 | 0.2 | 3 | 2 | 0.5 | 54 |
| FLOWERS102 | 6 | 4 | 0.4 | 1 | 0.4 | 6 | 5 | 0.9 | 62 |
| IMAGENET | 4 | 2 | 0.5 | 5 | 0.6 | 6 | 4 | 0.4 | 136 |
| CIFAR10 | 6 | 4 | 0.8 | 1 | 0.1 | 3 | 2 | 0.5 | 13 |
| CIFAR100 | 6 | 4 | 0.3 | 1 | 0.7 | 3 | 2 | 0.5 | 13 |
| SPORT8 | 3 | 2 | 0.3 | 1 | 0.2 | 3 | 2 | 0.8 | 43 |
| MIT67 | 3 | 2 | 0.6 | 1 | 0.8 | 4 | 2 | 0.6 | 48 |
| FLOWERS102 | 6 | 5 | 0.2 | 1 | 0.8 | 3 | 2 | 0.5 | 48 |

Table 4: Test accuracy (%) and memory consumption (MB) for variants of HNAG and RNAG after completing training. "BOHB" and "MOBO"indicates BO was used to optimise generator hyperparameters. "RNAG-D" is the best generator in [14] and "HNAG-RS" is the generator with hyperparameters randomly sampled from HNAG. Each table entry shows "mean (stdev)" of test accuracy (top row) and memory (bottom row) over 8 random samples. The best performance, separately for test accuracy and memory consumption, is highlighted in bold. Number of model parameters is limited to 4M for CIFAR10 and CIFAR100 and 6M for the other datasets.

|  | RNAG-BOHB | **HNAG-BOHB** | RNAG-MOBO | **HNAG-MOBO** | RNAG-D | **HNAG-RS** |
|---|---|---|---|---|---|---|
| CIFAR10 | 94.3(0.13) | **96.6(0.15)** | 94.0(0.26) | **96.6(0.07)** | 94.1(0.16) | 95.7(0.68) |
|  | 57.9(0.52) | 17.0(1.76) | 25.9(0.91) | **12.8(0.00)** | 44.2(1.29) | 53.9(40.6) |
| CIFAR100 | 73.0(0.50) | **79.3(0.31)** | 71.8(0.50) | 77.6(0.45) | 71.7(0.36) | 77.1(1.34) |
|  | 56.5(0.90) | 14.0(1.07) | 27.0(0.91) | **12.8(0.00)** | 43.5(1.23) | 72.6(40.2) |
| SPORT8 | 93.6(0.76) | 94.9(0.52) | 93.1(0.73) | **95.2(0.40)** | 93.6(0.99) | 93.2(1.55) |
|  | 101.9(1.18) | 121.9(13.1) | 57.8(1.07) | **43.1(0.00)** | 112.1(3.45) | 375.7(277) |
| MIT67 | 68.3(0.78) | **74.2(0.67)** | 66.9(1.46) | 73.5(0.56) | 66.7(0.54) | 72.5(1.38) |
|  | 156.1(6.40) | 54.1(2.50) | 56.7(0.59) | **48.1(0.00)** | 111.7(3.58) | 324.8(140) |
| FLOWERS102 | 95.7(0.38) | 97.9(0.18) | 94.9(0.53) | **98.1(0.19)** | 94.7(0.46) | 95.3(1.29) |
|  | 143.5(2.27) | 61.7(0.00) | 63.0(1.63) | **48.4(0.00)** | 111.6(3.35) | 211.2(140) |

## 4.2 BO Experiments

BOHB is used to find the optimal network generator hyperparameters in terms of the validation accuracy. We perform BOHB for 60 iterations. We use training budgets of $100, 200, 400$ epochs to evaluate architectures on SPORT8 and $30, 60, 120$ epochs on the other datasets. MOBO returns the Pareto front of generator hyperparameters for two objectives: validation accuracy and sample memory. For parallel MOBO, we start with 50 BOHB evaluation data and search for 30 iterations with a BO batch size of 8; at each BO iteration, the algorithm recommends 8 new points to be evaluated and updates the surrogate model with these new evaluations. We use a fixed training budget of 200 epochs to evaluate architectures suggested for SPORT8 and 60 epochs for the other datasets. For experiments with both BO methods, we only sample 1 architecture to evaluate the performance of a specific generator. We scale the batch size up to a maximum of 512 to fully utilise the GPU memory and adjust the initial learning rate via linear extrapolation from 0.025 for a batch size of 96 to 0.1 for a batch size of 512. The other network training set-up follows the complete training protocol.

The validation accuracy of the best generator configuration recommended by BOHB as well as the computation costs to complete the BOHB search for both our proposed hierarchical network generator (HNAG) and the Randomly Wired Networks Generator (RNAG) are shown in Table 2. HNAG improves over RNAG in terms of both accuracy and cost. The high rank correlation between architecture accuracies at subsequent budgets (see Fig. 5b in Appendix I) indicates that a good generator configuration remains good even when a new architecture is re-sampled and re-evaluated at a higher budget; this reconfirms the validity of our practice to assess the generator configuration performance with only one architecture sample during the BO search phase.

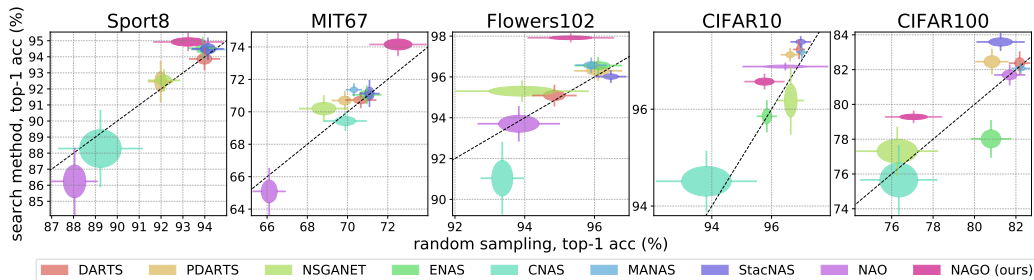

Figure 4: Comparison of various NAS methods (performance on y-axis) and random sampling (performance on x-axis) from their respective search spaces. Ellipse centres, ellipse edges and whisker ends represent the $\mu \pm \{0, \sigma/2, \sigma\}$, respectively (mean $\mu$, standard deviation $\sigma$). Methods above the diagonal line outperform the average architecture, and vice-versa. Note NAGO clearly achieves the largest improvement over naive random sampling than all other methods on all tasks. Results for all methods except ours (NAGO: HNAG-BOHB) were taken from public resources provided by [7]. While all high-performing competing methods use Cutout, DropPath and Auxiliary Towers, our method only uses Cutout. The other methods are: DARTS [1], PDARTS [2], NSGANET [50], ENAS [3], CNAS [51], MANAS [4], StacNAS [5], NAO [52].

The query data by MOBO over the two objectives, validation error and sample memory consumption are presented in Figure 3. Clearly, the Pareto front of HNAG dominates that of RNAG, showing that our proposed search space not only performs better but is also more memory efficient.

### 4.3  Analysis of the Optimal Hyperparameters

In Table 3, most optimal generators have much fewer nodes ($\leq 40$ nodes) than the graph-based generator in [14] (96 nodes) while still achieve better performance (Section 4.4). This shows that our hierarchical search space helps reduce the complexity/size of the architectures found. Interestingly, most datasets have the optimal solution with a single-node mid-level graph. We hypothesize this to be due to the low parameter count we enforced, which encourages the search to be conservative with the total number of nodes. Moreover, we see similar configurations appear to be optimal for different datasets, showing the promise of using transfer-learning to speed up the search in our future work.

### 4.4  Complete Training Results

We train (i) the best network generator hyper-parameters of both HNAG and RNAG found by BO; (ii) the default optimal RNAG setting (RNAG-D) recommended in [14]; and (iii) randomly sampled hyperparameters values for our HNAG (HNAG-RS) following the complete training protocol. HNAG clearly outperforms RNAG (Table 4). Moreover, in the multi-objective case, HNAG-MOBO is able to find models which are not only competitive in test accuracy but also very lightweight (i.e., consuming only $1/3$ of the memory when compared to RNAG-D).

Table 5: Test accuracy (%) of small networks ($\sim$6M paremeters) on IMAGENET. We train our HNAG-BOHB for 250 epochs similar to RandomWire-WS [14].

| Network | Top-1 acc. | Top-5 acc. | Params(M) |
|---|---|---|---|
| ShuffleNetV2 [53] | 74.9 | 92.2 | 7.4 |
| NASNet [54] | 74.0 | 91.6 | 5.3 |
| Amoeba [55] | 75.7 | 92.4 | 6.4 |
| PNAS [56] | 74.2 | 91.9 | 5.1 |
| DARTS [1] | 73.1 | 91.0 | 4.9 |
| XNAS [8] | 76.0 | — | 5.2 |
| RandWire-WS | 74.7 | 92.2 | 5.6 |
| HNAG-BOHB | **76.8** | **93.4** | 5.7 |

An interesting analysis is presented in Figure 4. This plot shows the relationship between randomly-sampled and method-provided architectures, and is thus able to separate the contribution of the search space from that of the search algorithm. Notably, not only does NAGO provide high accuracy, but also has the best relative improvements of all methods. It must be noted that while these methods train their networks using Cutout [57], DropPath [58] and Auxiliary Towers [13], *we only used Cutout*. DropPath and Auxiliary Towers could conceivably be used with our search space [6], although an effective and efficient implementation is non-trivial. Furthermore, the competitive performance of

one-shot NAS methods is largely due to the above-mentioned training techniques and their well-designed narrow search space [7]. Finally, while the number of parameters in other methods can vary quite a lot (the number of channels is fixed, regardless of the specific solution), NAGO dynamically computes the appropriate number of channels to preserve the given parameter count.

We also perform the search on IMAGENET. Due to resource constraints, we only run 10 iterations of BOHB with search budgets of $15, 30, 60$ epochs and train 2 sample networks from the best generator hyperparameters recommended by BOHB, following the complete training protocol in [14]. Although this is likely a sub-optimal configuration, it serves to validate our approach on large datasets such as IMAGENET. The mean network performance achieved by HNAG-BOHB approach outperforms [14] and other benchmarks (Table 5). Note that XNAS uses DropPath, Auxiliary Towers and AutoAugment [59] to boost performance.

## 5    Discussion and conclusion

We presented NAGO, a novel solution to the NAS problem. Due to its highly-expressive hierarchical, graph-based search space, together with its focus on optimizing a generator instead of a specific network, it significantly simplifies the search space and enables the use of more powerful global optimisation strategies. NAGO, as other sample-based NAS methods, requires more computation ($< 20$ GPU-days) than one-shot NAS methods ($< 2$ GPU-days). The later are geared towards fast solutions, but rely heavily on weight-sharing (with associated drawbacks [60]) and a less expressive search space with fixed macro-structure to achieve this speed-up, and tend to overfit to specific datasets (CIFAR), while under-performing on others (Figure 4). Additionally, while the architectures found by NAGO are already extremely competitive, the training protocol is not fully optimised: NAGO does not use DropPath or Auxiliary Towers—which have been used to significantly boost performance of one-shot NAS [7]—so additional accuracy gains are available, and we aim to transfer these protocols to our HNAG backbone. For future direction it would be interesting to consider a lifelong NAS setting or transfer learning, in which each new task can build on previous experience so that we can quickly get good results on large datasets, such as ImageNet. This can be more easily achieved with our novel search space HNAG as it allows us to deploy the existing transfer-learning BO works directly. In addition, the network generator hyperparameters define the global properties of the resultant architectures—network connectivity and operation types—and from these we can derive a high level understanding of the properties that make up a good architecture for a given task.

## 6    Broader Impact

As highlighted in [7], NAS literature has focused for a long time on achieving higher accuracies, no matter the source of improvement. This has lead to the widespread use of narrowly engineered search spaces, in which all considered architectures share the same human defined macro-structure. While this does lead to higher accuracies, it prevents those methods from ever finding truly novel architecture. This is detrimental both for the community, which has focused many works on marginally improving performance in a shallow pond, but also for the environment [61]. As NAS is undoubtedly computationally intensive, researchers have the moral obligation to make sure these resources are invested in meaningful pursuits: our flexible search space, based on hierarchical graphs, has the potential to find truly novel network paradigms, leading to significant changes in the way we design networks. It is worth mentioning that, as our search space if fundamentally different from previous ones, it is not trivial to use the well-optimised training techniques (e.g. DropPath, Auxiliary Towers, etc.) which are commonly used in the field. While transferring those techniques is viable, we do believe that our new search space will open up the development of novel training techniques.

We do however acknowledge that the computational costs of using our NAS approach are still relatively high - this may not be attractive to the industrial or academic user with limited resources. On the other hand, by converting NAS to a low-dimensional hyperparameter optimisation problem, we have significantly reduced the optimisation difficulty and opened up the chance of applying more optimisation techniques to NAS. Although only demonstrated with BOHB and MOBO in this work, we believe more query-efficient methods, such as BO works based on transfer learning [62, 63, 64, 65, 66, 67] can be deployed directly on our search space to further reduce the computation costs.

# 7 Acknowledgement

We thank our colleagues, especially Steven McDonagh, at Huawei Noah's Ark Lab (London) for their useful feedback. Binxin Ru was supported by the Clarendon Fund of University of Oxford.

## Footnotes

[1]The WS model cannot generate a single-node graph but the ER model can.

[2]For example, a 32-nodes graph has 496 possible connections. If we divide this into 4 subgraphs of 8 nodes, that number is $118 = 28 \times 4$ (within subgraphs) $+ 6$ (between subgraphs).

[3]Note modifying BOHB to also accommodate the multi-objective setting is an interesting future direction. One potential way is to do so is by selecting the Pareto set points at each budget to be evaluated for longer epochs during Successive Halving.

[4]Namely the 3 WS graph hyperparameters at top and bottom levels, $\boldsymbol{\theta}_{top} \in \mathbb{R}^3, \boldsymbol{\theta}_{bottom} \in \mathbb{R}^3$; and the 2 ER graph hyperparameters at mid level, $\boldsymbol{\theta}_{top} \in \mathbb{R}^2$.

[5]See Appendix F for a comparison with DARTS search space.

[6]We naively apply DropPath and Auxiliary Towers, following set-ups in [1], to re-train architectures from our hierarchical search space. These techniques lead to 0.54% increase in the average test accuracy 25 over 8 architecture samples on CIFAR10, leading to results competitive with the state-of-the-art.

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
