[Supplementary Material]

# Neural Architecture Generator Optimization
# (Appendix)

**Binxin Ru**
Machine Learning Research Group
University of Oxford, UK
robin@robots.ox.ac.uk

**Pedro M. Esperança**
Huawei Noah's Ark Lab
London, UK
pedro.esperanca@huawei.com

**Fabio M. Carlucci**
Huawei Noah's Ark Lab
London, UK
fabio.maria.carlucci@huawei.com

## A    Comparison of our hierarchical search space (HNAG) with previous ones

### A.1    Quantifying the expressiveness of the hierarchical search space (HNAG) against DARTS search space

The total number of possible graphs in our hierarchical search space is larger than

$$T = \underbrace{\left(\sum_{n=3}^{N_O} 2^{\phi(n)}\right)}_{\text{Operation-level}} \cdot \underbrace{\left(\sum_{n=1}^{N_C} 2^{\phi(n)}\right)}_{\text{Cell-level}} \cdot \underbrace{\left(\sum_{n=3}^{N_S} 2^{\phi(n)} \cdot M^n\right)}_{\text{Stage-level}}$$

where $\phi(n) = n(n+1)/2$ is the number of possible DAGs with $n$ nodes; $N_O$, $N_C$, $N_S$ are the maximum numbers of nodes in the operation-, cell- and stage-level graphs, respectively; and $M$ is the number of possible operations in the operation-level graph.

**NOTE:** This calculation does not include all variations due to the different merging possibilities for each node (addition and concatenation).

Concretely, for the setting implemented ($N_O = N_C = N_S = 10$, $M = 5$) we have $T_{\text{HNAG}} \approx 4.58 \times 10^{56}$. For comparison, the DARTS search space has $T_{\text{DARTS}} = 8^{14} \approx 4.40 \times 10^{12}$.

### A.2    Example architectures from HNAG search space

We show six sample architectures drawn from our proposed HNAG search space in Figure 1. It's evident that our hierarchical graph-based search space can generate a large variety of architectures. Note Figure 1 (a) corresponds to the optimal architecture proposed in [1], which is also contained in our search space.

### A.3    Illustrating difference from RNAG [1]

While RNAG is *flat* with 3 sequentially connected graphs, HNAG is *hierarchical* with 3 levels and each node in the higher level corresponds to a graph in the level below. Thus, the middle graph in RNAG does not correspond to the middle-level graph in HNAG. Our 3-level hierarchical structure is not only a generalisation which enables the creation of more complex architectures, but it also allows the creation of local clusters of operation units, which result in more memory efficient models or architectures with fewer nodes as shown in the main paper and Figure 1. Moreover, the HNAG

(a) $\boldsymbol{\theta}_{top} = (3, 2, 0.8), \boldsymbol{\theta}_{mid} = (1, 1.0),$ $\boldsymbol{\theta}_{bottom} = (32, 4, 0.75)$ (RNAG default)

(b) $\boldsymbol{\theta}_{top} = (8, 0.4, 5), \boldsymbol{\theta}_{mid} = (1, 0.7),$ $\boldsymbol{\theta}_{bottom} = (4, 2, 0.4)$

(c) $\boldsymbol{\theta}_{top} = (6, 0.4, 4), \boldsymbol{\theta}_{mid} = (1, 0.4),$ $\boldsymbol{\theta}_{bottom} = (6, 4, 0.4)$

(d) $\boldsymbol{\theta}_{top} = (6, 0.4, 4), \boldsymbol{\theta}_{mid} = (1, 0.4),$ $\boldsymbol{\theta}_{bottom} = (6, 5, 0.9)$

(e) $\boldsymbol{\theta}_{top} = (7, 0.9, 2), \boldsymbol{\theta}_{mid} = (5, 0.8),$ $\boldsymbol{\theta}_{bottom} = (6, 3, 0.6)$

(f) $\boldsymbol{\theta}_{top} = (4, 0.5, 2), \boldsymbol{\theta}_{mid} = (5, 0.6),$ $\boldsymbol{\theta}_{bottom} = (6, 4, 0.4)$

Figure 1: Our proposed HNAG search space contains a large diversity of architectures

top level provides diverse connections across different stages, leading to more flexible information flow than RNAG. Finally, nodes in RNAG only process features of fixed channel size and resolution within a stage while those in HNAG receive features with different channel sizes and resolution. To summarize, our proposed search space HNAG is significantly different from RNAG.

## B   Neural Architecture Generator Optimisation (NAGO) Algorithm

BO is a technique for optimizing a black-box function which is usually noisy and expensive to evaluate. The two key components of BO are: a statistical surrogate model which models the unknown objective; and an acquisition function which is optimized to recommend the next query location [2, 3]. Our NAGO algorithm deploys BO to optimise over the low-dimensional continuous search space of generator hyperparameters. In summary, NAGO trains the surrogate model from

Figure 2: Architecture sampled from RNAG. RNAG is flat but our HNAG is hierarchical.

---

**Algorithm 1** Network Architecture Generator Optimization

---

1: **Input:** Network generator $G$, BO surrogate model $p(f|\mathbf{\Theta}, D)$ and acquisition function $\alpha(\mathbf{\Theta}|D)$
2: **for** $t = 1$ **to** $T$ **do**
3:      Recommend $\{\mathbf{\Theta}_t^j\}_{j=1}^B = \arg\max \alpha_{t-1}(\mathbf{\Theta}|D)$
4:      **for** $j = 1$ **to** $B$ **in parallel do**
5:          Sample an architecture from $G(\mathbf{\Theta}_t^j)$ and evaluate its validation performance $f_t^j$
6:      **end for**
7:      Update $D$ and thus $p(f|\mathbf{\Theta}, D)$ with $\{\mathbf{\Theta}_t^j, f_t^j\}_{j=1}^B$
8: **end for**
9: Obtain the best performing $\mathbf{\Theta}^*$ or the Pareto set $\mathbf{\Theta}^*$
10: Sample 8 architectures from $G(\mathbf{\Theta}^*)$, train them to completion and report their test performance.

---

query data and uses it to build an acquisition function which trades off exploitation and exploration. At each iteration, NAGO recommends $B$ new generator hyperparameter values by maximizing the acquisition function and updates the surrogate after evaluating these $B$ points. The full algorithm of NAGO is presented in Algorithm 1.

## C Hyperparameters of BO algorithms in NAGO

For experiments with both BO methods, we only sample 1 architecture to evaluate the performance of a specific generator. We scale the batch size up to a maximum of 512 to fully utilise the GPU memory and adjust the initial learning rate via linear extrapolation from 0.025 for a batch size of 96 to 0.1 for a batch size of 512. The other network training set-up follows the complete training protocol described from line 202 to 208 in Section 4.

### C.1 BOHB hyperparameters and set-up

We use the released code of BOHB [1]. We perform BOHB for 60 iterations with its hyperparamter $\eta = 2$. All the other BOHB hyperparameters follow the default setting in [4]. We use training budgets of $100, 200, 400$ epochs to evaluate architectures on SPORT8 and $30, 60, 120$ epochs on the other datasets.

### C.2 Hyperparameters of MOBO and its heteroscedastic Bayesian Neural Network Surrogate

MOBO returns the Pareto front of generator hyperparameters for two objectives: validation accuracy and sample memory. For parallel MOBO, we start with 50 initial data from BOHB queries and search for 30 iterations with a BO batch size of 8; at each BO iteration, the algorithm recommends 8 new

points to be evaluated and updates the surrogate model with these new evaluations. We use a fixed training budget of 200 epochs to evaluate architectures suggested for SPORT8 and 60 epochs for the other datasets.

Our Bayesian neural network surrogate is a 3-layer fully connected network with 10 neurons for each layer and two final outputs: predicted validation accuracy and heteroscedastic noise variance. For sampling network weights, we perform $5 \times |D|$ SGHMC steps as burn-in, followed by $10 \times 100$ sampling steps (retaining every 10th sample). We use a total of 100 samples of $w_f$ to approximate the integration in Equation (1) in the main paper. All the other hyperparameters of SGHMC follow the default setting in [5]. We implemented this surrogate by modifying the code of [5] [2].

## D   Local Penalisation for Batch Bayesian Optimization

We adopt the hard local penalization method proposed in [6] to collect a batch of new generator configurations which are then evaluated in parallel. The method sequentially selects a batch of $B$ new configurations by repeatedly applying a hard local penalizer function on the selected points (Algorithm 2).

---

**Algorithm 2** Local Penalisation

---

1: **Input:** BO surrogate model $p(f|\Theta, D)$ and acquisition function $\alpha(\Theta|D)$, BO batch size $B$, Local penalization function $\phi(\Theta|\Theta^j)$
2: **Output:** The batch of new configurations $\mathcal{B} = \{\Theta^j\}_{j=1}^B$
3: $\Theta^1 = \arg\max \alpha(\Theta|D)$ and $\mathcal{B} = \{\Theta^1\}$
4: **for** $j = 2, \ldots, B$ **do**
5: $\quad \Theta^j = \arg\max \left( \alpha(\Theta|D) \prod_{i=1}^{j-1} \phi(\Theta|\Theta^i) \right)$
6: $\quad \mathcal{B} \leftarrow \mathcal{B} \cup \Theta^j$
7: **end for**

---

The hard penalisation function is defined as:

$$\phi(\Theta|\Theta^j) = \min \left\{ \frac{L\|\Theta - \Theta^j\|}{|\mu(f|\Theta, D) - M| + \sigma(f|\Theta, D)}, 1 \right\}$$

where $M$ is the best objective value observed so far, $L = \max_{\Theta} \|\bigtriangledown \mu(f|\Theta, D)\|$ is the approximated Lipschitz constant of the objective function, and $\mu(f|\Theta, D)$ and $\sigma(f|\Theta, D)$ are predictive posterior mean and standard deviation of the BO surrogate model.

## E   Search Range of Generator Hyperparameters

For our Hierarchical Neural Architecture Generator (HNAG), the ranges over which the generator hyperparameters are searched are defined as:

**Hyperparameters of the top-level and bottom-level Watts–Strogatz graphs**

- The number of nodes in the graph $N_t, N_b \in [3, 10]$
- The number of nearest neighbors to which each node is connected in ring topology $K_t, K_t \in [2, 5]$
- the probability of rewiring each edge $P_t, P_b \in [0.1, 0.9]$

**Hyperparameters of the Mid-level Erdős–Rényi graph**

- The number of nodes in the graph $N_m \in [1, 10]$
- the probability of edge creation $P_m \in [0.1, 0.9]$

Figure 3: Memory consumption histograms of 300 sample architectures from HNAG, RNAG and DARTS search spaces for small-image $32 \times 32 \times 3$ and large-image $224 \times 224 \times 3$ datasets. Our HNAG search space can generate architectures with a wider range of memory consumption, especially for the large-image data.

For the Randomly Wired Neural Architecture Generator (RNAG), the hyperparameter ranges are:

**Hyperparameters of the Watts–Strogatz graphs in 1st, 2nd and 3rd stages**

- The number of nodes in the graph $N_1, N_2, N_3 \in [10, 40]$

- The number of nearest neighbors to which each node is connected in ring topology $K_1, K_2, K_3 \in [2, 9]$

- the probability of rewiring each edge $P_1, P_2, P_3 \in [0.1, 0.9]$

Note that although HNAG has a smaller range for the number of nodes in each graph $N$ than RNAG does, it actually can lead to a much larger range of total number of nodes in an architectures ($[9, 1000]$) than that of RNAG ($[30, 120]$).

## F    Memory Consumption Range of Architectures from Different Search Space

Our hierarchical graph-based search space can generate architectures with a wider range of memory consumption than those of RNAG and DARTS. We draw 300 sample architectures from the search spaces of HNAG, RNAG and DARTS and evaluate their memory consumption per image. The histogram for results on small-image data and large-image data are shown in Figure 3. It is evident that our proposed search space is much wider than both RNAG and DARTS in terms of the memory consumption.

## G    Performance of randomly sampled network generator hyperparameters during BO search phase

In Figure 4, we evaluate the test performance of 50 randomly sampled network generator hyperparameters for CIFAR10. For each generator hyperparameter value, we sample 8 neural network architectures and train them for 60 epochs following the protocol of the BO search phase: we scale the batch size up to a maximum of 512 to fully utilise the GPU memory and adjust the initial learning rate via linear extrapolation from $0.025$ for a batch size of $96$ to $0.1$ for a batch size of $512$. The observations we made on Figure 2 in the main paper also hold for Figure 4.

Figure 4: The mean and standard deviation of test error vs. memory consumption and training time per epoch achieved by $50$ random generator hyperparameters for CIFAR10 after training for $60$ epoches following BO search phase protocol.

# H BOHB results on searching more generator hyperparameters

## H.1 Include hyperparameters controlling merge options and node operations

We also perform BOHB on an expanded search space $\boldsymbol{\Theta}_{augV1}$ which includes not only the original space of the three random graph model hyperparameters $\boldsymbol{\Theta}_{origin} = [\boldsymbol{\theta}_{top}, \boldsymbol{\theta}_{mid}, \boldsymbol{\theta}_{bottom}]$ but also hyperparameters controlling the merge options and node operations $\boldsymbol{\theta}_M$ and $\boldsymbol{\theta}_{op}$. Specifically, $\boldsymbol{\theta}_M$ defines the probability of choosing weighted sum or concatenation when merging multiple inputs at a node. $\boldsymbol{\theta}_M$ defines the probability of choosing a specific operation among (conv1 $\times$ 1, conv3 $\times$ 3, conv5 $\times$ 5, pool3 $\times$ 3 and pool5 $\times$ 5) for each node in the bottom-level graph. The stage ratio and channel ratio are still fixed to $\boldsymbol{\theta}_S = [0.33, 0.33, 0.33]$ and $\boldsymbol{\theta}_C = [1:2:4]$ following [1]. Therefore, the expanded search space is $\boldsymbol{\Theta}_{augV1} = [\boldsymbol{\theta}_{top}, \boldsymbol{\theta}_{mid}, \boldsymbol{\theta}_{bottom}, \boldsymbol{\theta}_M, \boldsymbol{\theta}_{op}]$.

As seen in Table 1, the best validation accuracies achieved by HNAG-AugV1 are lower than that by HNAG for all the datasets. This result is counter-intuitive as $\boldsymbol{\Theta}_{origin} \subset \boldsymbol{\Theta}_{augV1}$ and thus searching on $\boldsymbol{\Theta}_{expanded}$ should lead to better or at least equal performance as on $\boldsymbol{\Theta}_{origin}$. Yet, this result can be explained by the follwoing two reasons:

1) the significant increase in optimisation difficulty. The search dimensionality of $\boldsymbol{\Theta}_{augV1}$ is almost twice that of $\boldsymbol{\Theta}_{origin}$, which significantly increases the difficulty of BOHB in finding the global optimum [3]. Thus, given similar search budget, BOHB is more likely to find a hyperparameter near the global optimum or a better local optimum in the space $\boldsymbol{\Theta}_{origin}$ than in the expanded space $\boldsymbol{\Theta}_{augV1}$.

2) the marginal gain in expanding the search space. [1] empirically demonstrate that the wiring pattern in a architecture plays a much more important role than the operation choices. Our result in Table 1 confirms this observation; namely, after finding the good wiring, changing the operations only lead to small perturbation on the generator performance. Putting this in the context of generator optimisation, it means that the wiring hyparameters $\boldsymbol{\Theta}_{origin}$ determines the region where the global optimum locates and the hyperparameters controlling the operation and merge options only perturb the exact location of the global optimum to a small extent.

Combing the above two factors, we attribute the worse validation performance for HNAG-AugV1V1 to the fact that the increase in optimisation difficulty far outweights the gain in expanding the search space.

Table 1: Validation accuracy (%) and search cost (GPU days) for BOHB results. The accuracy reported is obtained in the BOHB search setting which uses large batch sizes based on GPU machine memory and trains the network sample for 400 epochs for SPORT8 and 120 epochs for the other datasets. The search space of HNAG-AugV1 is $\Theta_{augV1} = [\theta_{top}, \theta_{mid}, \theta_{bottom}, \theta_M, \theta_{op}] \in \mathbb{R}^{15}$ while that of HNAG is $\Theta_{orign} = [\theta_{top}, \theta_{mid}, \theta_{bottom}] \in \mathbb{R}^8$.

| | HNAG-AugV1 | | HNAG | |
| --- | --- | --- | --- | --- |
| | Accuracy | Cost | Accuracy | Cost |
| CIFAR10 | 94.7 | 19.2 | **95.6** | **12.8** |
| CIFAR100 | 74.5 | 21.3 | **77.2** | **10.4** |
| SPORT8 | 94.0 | 26.1 | **95.3** | **17.6** |
| MIT67 | 68.8 | 33.3 | **71.8** | **20.0** |
| FLOWERS102 | 91.0 | 14.4 | **93.3** | **10.6** |

Table 2: Validation accuracy (%) during search and network training time per epoch (seconds). The accuracy reported is obtained in the BOHB search setting which uses large batch sizes based on GPU machine memory and trains the network sample for 400 epochs for SPORT8 and 120 epochs for the other datasets. The search space of HNAG-AugV2 is $\Theta_{augv2} = [\theta_{top}, \theta_{mid}, \theta_{bottom}, \theta_S, \theta_C] \in \mathbb{R}^{14}$ while that of HNAG is $\Theta_{orign} = [\theta_{top}, \theta_{mid}, \theta_{bottom}] \in \mathbb{R}^8$.

| | HNAG-AugV2 | | HNAG | |
| --- | --- | --- | --- | --- |
| | Accuracy | Mean(Max) Time | Accuracy | Mean(Max) Time |
| CIFAR10 | **95.7** | 99.3(998) | 95.6 | **54.9(246)** |
| CIFAR100 | **77.5** | 82.2(711) | 77.2 | **43.0(216)** |
| SPORT8 | **95.9** | **20.6(93.6)** | 95.3 | 22.8(37.0) |
| MIT67 | **72.0** | 130(1056) | 71.8 | **85.4(291)** |
| FLOWERS102 | **93.3** | 58.4(397) | 93.3 | **45.4(105)** |

## H.2 Include hyperparameters controlling stage ratios and channel ratios

We then perform similar experiments like above but instead optimise the hyperparameters controlling the stage ratios $\theta_S$ and channel ratios $\theta_C$ while keeping $\theta_M$ and $\theta_{op}$ fixed. The experimental results are shown in Table 2. While there is a marginal increase in performance, the worst case train time substantially increases due to extreme stage and channel ratios. So, while the number of architectures sampled stays the same, the computational cost increases due to more lengthy training. We had observed this effect during preliminary experiments on CIFAR10 and thus decided to fix $\theta_S$ and $\theta_C$ to standard values in order to obtain competitive results at a reasonable cost. Nonetheless, even with such constrains on the search space, our HNAG is still much more expressive than most NAS search spaces.

# I BOHB samples

Here we show the BOHB query results on the generator hyperparameters for the case of CIFAR10. We use three training budgets in BOHB: 30 (green), 60 (orange) and 120 (blue) epochs. In Figure 5, the top subplot shows the validation error for the three budgets over time. Query data for different budget are mostly well separated. The bottom subplot shows the spearman rank correlation coefficients $\rho_{spearman} \in [-1, 1]$ of the validation errors between different budgets. It's evident that the $\rho_{spearman}$ between data of 60 epochs and those of 120 epochs are quite high (0.82), indicating that good hyperparameters found in the budget of 60 epochs will remain good when being evaluated with 120 epochs. This motivates our to only a fixed budget of 60 epochs for evaluating all the hyperparameter samples in the multi-objective BO setting.

(a) Validation error for different budgets over time     (b) Rank correlation of validation errors across budgets

Figure 5: BOHB query data across different budgets for HNAG on CIFAR10

## Footnotes

[1]Available at `https://github.com/automl/HpBandSter`

[2]Available at `https://github.com/automl/pybnn`

[3]To optimize a function to within $\epsilon$ distance from the global optimum using random search, the expected number of iterations required is $\mathcal{O}(\epsilon^{-d})$ [7]