[Reviews · NeurIPS 2020]

Review 1

Summary and Contributions: The paper proposes to summarize network architectures into a graph generator parameterized by a small number of variables, and to explicitly optimize the generator’s configuration to find better architectures (using Bayesian Optimization). Different from the commonly used cell-based search spaces, the graph generative process is defined over the entire network in a hierarchical manner. Experiments show promising improvements on image classification tasks, both on small datasets (e.g., CIFAR, Flowers) and on ImageNet.

Strengths: * The paper confirms the interesting phenomenon that an architecture’s performance is quite explainable by the graph generator from which it was sampled. * The idea of optimizing the graph generator is a natural yet meaningful extension of [a]. The fact that the graph generators are of small degrees of freedom enables the usage of well understood optimization tools (e.g., BO) to search for complex architectures. * The paper is well written and well-motivated. It has also provided convincing ablation studies for several design choices made. [a] Xie et al. "Exploring randomly wired neural networks for image recognition." ICCV 2019.

Weaknesses: * In experiments the authors fixed \theta_S and \theta_C as constants (Section 4, L185-186), two of the most critical factors that control the global resource allocation of the networks. This makes the search space substantially more restricted than what they were claimed to be in Sections (1-3). I would find the message more appealing if \theta_S and \theta_C are searched, as both are likely going to strongly affect the network’s overall performance. With the current settings, it remains unclear whether BO is able to handle the enlarged search space with reasonable sample efficiency. * The authors claim the hierarchical graph-based search space as a major contribution, but similar ideas have been studied by quite a few prior works, e.g., [b, c, d], and none of them were mentioned. These need to be discussed to accurately reflect one of the primary contributions of this paper, namely using hierarchical representation for network generators (rather than the hierarchical representation itself). [b] Liu et al. "Hierarchical representations for efficient architecture search." ICLR 2018. [c] Liu et al. "Auto-DeepLab: Hierarchical Neural Architecture Search for Semantic Image Segmentation." CVPR 2019. [d] Tan et al. "MnasNet: Platform-Aware Neural Architecture Search for Mobile" CVPR 2019. ================ update after reading the rebuttal ================ I'd like to thank the authors for providing the additional results with searchable \theta_S and \theta_C. I believe including these results in the main paper can substantially strengthen the work, as it is important to ensure consistency between the method description/claims and the experimental setup.

Correctness: Yes.

Clarity: Yes. The paper is very well written.

Relation to Prior Work: No. Please refer to my comments on hierarchical representations in the "weaknesses" section.

Reproducibility: Yes

Additional Feedback: The paper focused on accuracy-params tradeoff for its main results. It would be interesting to consider FLOPs and/or inference latency, as they are usually more relevant in practice as resource constraints.


Review 2

Summary and Contributions: This work proposes NAGO, a neural architecture search approach based on neural architecture operator optimization. Similar to the previous work [1], it focuses on searching for an architecture generator instead of a specific architecture. It expands [1] by proposing a hierarchical search space to replace a flat search space proposed in [1], and proposing a Bayesian optimization (BO) based search strategy to replace the random search strategy used in [1]. The authors demonstrated the effectiveness of the proposed approach on six benchmark datasets and show that their approach is able to find extremely lightweight yet highly competitive models. [1] Exploring Randomly Wired Neural Networks for Image Recognition, ICCV 2019.

Strengths: Casting NAS as a problem of finding the optimal network generator is a natural extension of the pioneer work proposed in [1]. The formulation of optimizing neural network generators with BO is reasonable. Moreover, this work has conducted very solid and comprehensive evaluations on their proposed approach, demonstrating the expressiveness of its proposed search space, and its strong BO-based architecture search results. This paper could raise more awareness of the design of the search space which is important for the community.

Weaknesses: I have two concerns. The first concern lies at the design of the proposed hierarchical graph-based search space (HNAG). This search space is hand designed with specific types of network generators at different hierarchies (top and bottom levels use WS random network generator and middle level uses ER random network generator). It is not clear to me what are the rationales and insights behind such design. Why the hierarchical search space contains 3 levels? Why WS is selected for top and bottom levels and ER is selected for middle level? Since the proposed HNAG is the key innovation of this work, it is highly desired for the authors to explain its design rationale and insights. The second concern lies at some of its evaluation methodology and results. For experiments with BO, the authors only sampled 1 architecture to evaluate the performance of a specific generator. Using only one sample for evaluating the performance of the generator is questionable given that the variation in Figure 2 is not that small for many of the network generator configurations. In Table 3, as pinpointed by the authors, most datasets have the optimal solution with a single-node mid-level graph. But I am not fully convinced by the hypothesis provided by the authors. In Figure 5, among 5 datasets, only one dataset (MIT67) shows the superiority of the proposed hierarchical graph-based search space (HNAG). This raises concerns on whether the proposed HNAG is truly effective. It would be more convincing if the authors could also add DropPath and Auxiliary Tower and compare the results again. Another way is to compare the test accuracy given a certain number of epochs. After reading the rebuttal, my concerns are addressed.

Correctness: This paper is technically correct.

Clarity: The paper is clear and well written.

Relation to Prior Work: Yes. The paper has a clear discussion on its relation to prior work.

Reproducibility: Yes

Additional Feedback: Provide rationales and insights behind the design of the proposed hierarchical graph-based search space (HNAG). Sample more architectures to evaluate the performance of a specific generator during BO experiments. Add DropPath and Auxiliary Tower and compare the results again in Figure 5.


Review 3

Summary and Contributions: ----------------------- Update after author response ---------------------------------- The authors addressed most of my concerns/questions. The authors showed results on a larger search space (with learnable stage ratios), which worked reasonable well (of course at the cost of much longer training time). While still some other design choices could be optimized, I do think this is an interesting and novel approach that could open up many future research and advance the field of NAS. Thus I think this paper should be accepted and I'm keeping my rating. ------------------------------------------------------------------------------------------------- The authors identify a major problem in current neural architecture search (NAS) research: many methods employ over-engineered search spaces, which makes it by design impossible to find truly novel architectures. To overcome this drawback, the authors propose a new perspective to NAS: to search for neural architecture generators rather than searching for a single best architectures within a prespecified search space. This comes with two major advantages: (i) a much richer space of architectures can be explored and (ii) generators have - in contrast to architectures - a low-dimensional parameterization, allowing for more efficient optimization. This low-dimensional parameterization allows the authors to use Bayesian Optimization to optimize the generators, not just with respect to high-performing architectures but also with respect to multiple objectives. The authors furthermore propose a hierarchical search space allowing for a more efficient search in a richer space. The proposed framework is thoroughly evaluated and ablated on various image recognition tasks (CIFAR10, CIFAR100, ImageNet, Sport8, MIT67, Flowers102).

Strengths: + novel approach to NAS based on neural architecture generator, overcoming major drawbacks of current methods + the proposed framework open ups a new research direction, with many potential follow-up work + thorough, fair experimental evaluation + very well written paper

Weaknesses: I do not have any major concerns, a few nitpicks are: - there are still a few manual design choices (such as the number of hierarchies, stages, graph generators per hierarchy, how tensors are merged,...) - from a purely practical/application point of view, the proposed method lacks behind in state-of-the-art performance since methods such as DropPath (which are basically specifically designed for well-known search spaces) do not easily transfer to the architectures generated by this work

Correctness: All claims made are correct, the empirical methodology is fair and correct.

Clarity: The paper is well structured and written, the proposed method is well motivated and introduced.

Relation to Prior Work: Prior work is to a large extent thoroughly discussed and set in contrast to the proposed work. The authors should add [1], which is also discussing the design search spaces, however in a more manual manner. [2] is also investigating different search spaces. Furthermore, the authors might want to consider mentioning work on multi-objective/constraint NAS such as [3,4], as the authors do also conduct multi-objective optimization. [1] Designing Network Design Spaces, Radosavovic et al., CVPR 2020 [2] Can weight sharing outperform random architecture search? An investigation with TuNAS, Bender et al., CVPR 2020 [3] Elsken et al., Efficient Multi-objective Neural Architecture Search via Lamarckian Evolution, ICLR 2019 [4] Cai et al., ProxylessNAS, ICLR 2019

Reproducibility: Yes

Additional Feedback: Lines 80-85: Are the number of channels increased to match the highest number? Did the authors experiment with other methods for matching dimensions, e.g., strided convolutions rather than pooling? Lines 60-63: It would be interesting to explore more options for the models chosen for each hierarchy. Ideally, the choice of model for each hierarchy could also be optimized. Figure 5 is a little unclear to me. Is there a fixed, underlying search space across all methods from which random samples are drawn, and these samples are then compared to models provided by the methods run on exactly this space? Or is every method run on the space proposed in the original paper? Also, for NAGO, do you compare samples from the optimized generator vs. samples from a not-optimized generator, or what exactly is the baseline here? Please explain the ad-hoc choice for the generators for different hierarchies (WS for top/bottom, ER for middle). It would be helpful to have a more detailed explanation on how to combine BOHB and MOBO, this seems not so straightforward to me. A few more baselines that come to my mind that would be interesting to see: 1) a method not using multi-fidelity optimization (such as vanilla BO) to evaluate the impact of multi-fidelity optimization 2) some evolution-based method since such methods are often used in NAS

[Author Response · NeurIPS 2020]

Table 1: Results on validation accuracy (%) during search and network training time per epoch (seconds).

| Metric | Experiment | CIFAR10 | CIFAR100 | MIT67 | FLOWER102 | SPORT |
|---|---|---|---|---|---|---|
| Search Validation Accuracy | $\theta_G$ | 95.6 | 77.2 | 71.8 | 93.3 | 95.3 |
| | $(\theta_G, \theta_S, \theta_C)$ | 95.7 | 77.5 | 72.0 | 93.3 | 95.9 |
| Mean(Max) Train Time | $\theta_G$ | 54.9(246) | 43.0(216) | 85.4(291) | 45.4(105) | 22.8(37.0) |
| | $(\theta_G, \theta_S, \theta_C)$ | 99.3(998) | 82.2(711) | 130(1056) | 58.4(397) | 20.6(93.6) |

We thank the reviewers for their positive comments and suggestions. As R1 and R3 note, our main contribution is a
new paradigm for NAS which, by focusing on network generators instead of single architectures, enables us to explore
a much larger search space but at much lower search dimensions. R1,R2,R3 agree this work presents a meaningful
extension to [14], has very solid and comprehensive evaluations, has strong BO-based search results, and is well written.
*Please refer to the main paper for [1], [7], [14].*

**R1. Search for stage ratio $\theta_S$ and channel ratio $\theta_C$** Results for experiments with learnable $\theta_S$ and $\theta_C$ are in Table 1.
While there is a marginal increase in performance, the worst case train time substantially increases due to extreme stage
and channel ratios. So, while the number of architectures sampled stays the same, the computational cost increases due
to more lengthy training. We had observed this effect during preliminary experiments on CIFAR10 and decided to fix
$\theta_S$ and $\theta_C$ to standard values in order to obtain competitive results at a reasonable cost. Nonetheless, even with such
constrains on the search space, our HNAG is still much more expressive than most NAS search spaces.

**R2,R3. Why WS and ER? Why 3 levels?** We choose WS as it was shown to offer the best performances [14]. The
middle level graph was modelled with ER as WS doesn't allow single node graphs: we wanted the flexibility to reduce
our search space to 2 levels and represent a DARTS-like architecture. Indeed HNAG is designed to be able to emulate
existing search spaces while also exploring potentially better new ones. As most existing NAS methods require 2 levels
to represent, we decided to have the option of falling back to 2 (when the mid-layer becomes single node) but allow for
3 levels, enabling the creation of local clusters of operation units, which can result in more memory efficient models.

**R2,R3. Fig.5** Fig. 5 compares random sampling (x-axis) vs NAS (y-axis) method performance. It is intended to
showcase which methods find a better architecture than the average one [7]. Every method is run with its original code
and search space. For NAGO the baseline is obtained by randomly sampling non-optimised generators. NAGO clearly
achieves the largest improvement over naive random sampling than all other methods on all tasks. Looking at the y-axis,
NAGO convincingly outperforms other NAS approaches on not only MIT67, but also Flowers102 and Sport8.

**R2. DropPath and Auxiliary Tower(AT).** We naively apply DropPath and AT, following set-ups in [1], to re-train
architectures from our hierarchical search space. These techniques lead to $0.54\%$ increase in the average test accuracy
over 8 architecture samples on CIFAR10, leading to results competitive with the state-of-the-art. However, as highlighted
in line 265 and acknowledged by R3, more efforts are needed to effectively adapt these techniques onto our new search
space, which is fundamentally different from existing NAS spaces. Note that training DARTS' best architecture without
these techniques leads to similar results as our NAGO (both 96.6 - see [7] Fig.3).

**R2. Search phase set-up.** Using 1 architecture sample during BO search is a reasonable compromise for the following
reasons: 1) while the variance of generator hyperparameters are not always small, the top performing generators (those
we are interested in) mostly have very tight performance variances (Fig.2 in paper and Fig. 4 in App.); 2) in BOHB
search, we resample architectures from the same generator hyperparameter when evaluated at different budgets. The
rank correlation between architecture accuracies at subsequent budgets is quite high (>0.82 - see Fig.5b in App.),
indicating that a good generator remains good even when a new architecture is reevaluated at a higher budget.

**R1. Optimising for memory vs FLOPS.** Our choice was to focus on model optimization for low-memory devices,
but we agree with R1 that FLOPs and inference times are also important metrics. Our multi-objective framework can be
easily adapted (by swapping the 2nd objective) to consider these objectives and we will investigate this in future work.

**R1. Hierarchical works** We thank the reviewer for suggesting these works; we have added a discussion regarding
them. Our main search space contribution is not with it being hierarchical per-se, but rather lies in a) its expressiveness
(orders of magnitude higher than competing methods), b) its representational compactness and c) its stochasticity.

**R3. Combine BOHB and MOBO.** We agree that it's not trivial to combine multi-fidelity BO with multi-objective BO,
especially in the BOHB framework. A potential direction is to select the Pareto set points, instead of points with highest
EI value, at each budget to be evaluated for longer epochs during Successive Halving. Added discussion to the paper.

**R3. Merging different channel sizes** Yes, the number of channels is increased to match the highest number. We only
tried pooling, but strided convolutions are a promising alternative.

**R3. References** We thank R3 for pointing out relevant related work, which we now reference in the paper.

[Meta-Review · NeurIPS 2020]

This paper initially got a borderline recommendation (6,5,7). The reviewers agree that this paper gives interesting findings and the idea is new -- it targets at how to generate search space. However, reviewers have some questions on the experiment results. The authors give good response and address these questions well. The ratings are increased to 7,7,6. All the reviewers give positive recommendation. AC agrees.